# Sparse Activations with Correlated Weights in Cortex-Inspired Neural Networks

Chanwoo Chun[1], Daniel D. Lee[2]

[1]Weill Cornell Medical College of Cornell University, [2]Cornell Tech

cc2465@cornell.edu, ddl46@cornell.edu

Although sparse activations are commonly seen in cortical brain circuits, the computational benefits of sparse activations are not well understood for machine learning. Recent neural network Gaussian Process models have incorporated sparsity in infinitely-wide neural network architectures, but these models result in Gram matrices that approach the identity matrix with increasing sparsity. This collapse of input pattern similarities in the network representation is due to the use of independent weight vectors in the models. In this work, we show how weak correlations in the weights can counter this effect. Correlations in the effective weights are introduced using a convolutional model, similar to the neural structure of lateral connections in the cortex. We show how to theoretically compute the properties of infinitely-wide networks with sparse, correlated weights and with rectified linear outputs. In particular, we demonstrate how the generalization performance of these sparse networks improves by introducing these correlations. We also show how to compute the optimal degree of correlations that result in the best-performing deep networks.

## 1. Introduction

Sparse neural activations are observed in both biological and artificial neural networks. For example, in the cortex, less than 10 % of the neurons are active at a given moment, yet cortical circuits are involved in amazingly complex sensory and motor computations [1]. This is true not only for mammalian brains but universally across the animal kingdom including the central nervous systems of insects. To understand the computational role of sparsity, Olshausen and Field [2] used artificial neural networks to show that introducing a sparsity-inducing regularizer, i.e. $L_1$ penalty, gives rise to receptive fields similar to those observed in the visual cortex. It has also been demonstrated that sparsity increases robustness in classification tasks [3]. More recently, the presence of sparse activation has been observed in high-performance neural networks such as AlexNet, LeNet, and various models of Transformers, even without explicit regularization for sparsity [4–9].

The sparsity in these neural network models is induced by a rectified linear unit (ReLU) with a large negative pre-activation bias that is implicitly learned when optimizing the error performance. In recent work investigating the emergence of sparsity in Transformer architectures, Li et al. [9] proved that single-step gradient descent from random initial weights increases the threshold of rectification by ReLU in a single hidden layer neural network, thereby increasing the sparsity. Other theoretical works [7, 10] prove that sparsely active neural networks can be learned to have an approximation power comparable to that of the denser counterparts, providing computationally efficient alternatives to the dense models.

Interestingly, harnessing the power of sparsity does not require tuning the sparse representation to the input data. Using infinitely wide random neural networks with rectified units, Babadi and Sompolinsky [3], Cho and Saul [11], Chun and Lee [12], Xie et al. [13] show wide-ranging performance benefits of data-independent sparsity. In particular, sparse and shallow networks are observed to have performances that are comparable to, if not better than, dense and deep counterparts. Although the widths of these networks are assumed to be infinitely wide to aid theoretical computation, they are shown to well-approximate the behavior of large finite networks. However,

First Conference on Parsimony and Learning (CPAL 2024).

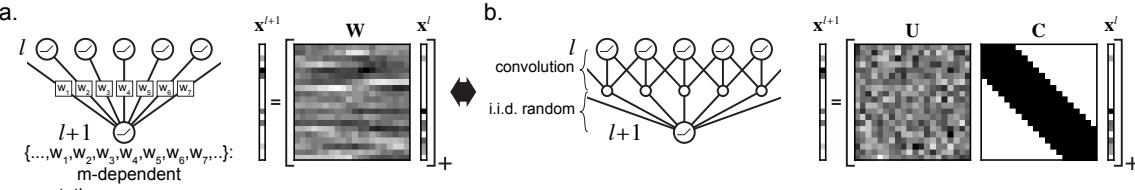

Figure 1: Model for correlated weights. (a) The correlated weights form an $m$-dependent stationary sequence. (b) The $m$-dependent weight matrix $\mathbf{W}$ factorizes into i.i.d. random matrix $\mathbf{U}$ and banded Toeplitz matrix $\mathbf{C}$, forming an intermediate convolutional layer. The models in (a) and (b) are equivalent. In NNGP, these weights are never explicitly sampled.

in these models, sparsity makes the neural representations of any pair of inputs more dissimilar, an effect that becomes amplified with more hidden layers. Therefore, sparse random models do not perform well with deep architectures [12]. Currently, there is little theoretical explanation of how deep network models benefit from sparsity. Here, we aim to bridge this gap by introducing a sparse model with weight correlations.

Inspired by the convolutional structure of cortical circuits, we introduce a mechanism for countering the dissimilating effect of sparsity by correlating the random weights. In the literature on the neural network Gaussian process (NNGP) and the mean-field theory of neural networks, the weights have been typically assumed to be independent so understanding the effect of correlated weights is currently limited. Most relevantly, however, Martí et al. [14] shows that symmetrically correlated weights can slow down dynamics in random recurrent networks, using a correlation and model structure different from the one proposed here.

### 1.1. Summary of contribution

We present a novel NNGP kernel formulation for correlated weights and empirically demonstrate that correlated weights enhance generalization performance in the sparse regime. This enhancement in performance is theoretically elucidated by leveraging recent advancements in generalization theory [15]. Additionally, we introduce a formula for calculating the optimal weight correlation at a given sparsity level.

## 2. Review of sparse NNGP kernel with independent weights

Here we consider ReLU-induced sparsity in a random neural network [12, 13], where the post-activation of $i^{th}$ neuron of an intermediate layer $l$ is given by

$$x_j^l = \left[h_j^l - b(\mathbf{x}^{l-1})\right]_+ \qquad h_j^l = \sum_i^{n^{l-1}} w_{ji}^{l-1} x_i^{l-1} \tag{1}$$

where $w_{ji}$ is a weight, and $b(\mathbf{x}^{l-1})$ is the bias that depends on the post-activation values of the previous layer. $[\cdot]_+$ denotes linear rectification. We refer to $h^l$ as the pre-activation. The first hidden layer $l = 1$ value is directly dependent on the input vector $\mathbf{x}$, whose elements are denoted $x_i$ (dropped the 0 superscript that indicates the layer number):

$$x_j^{l=1} = \left[h_j^{l=1} - b(\mathbf{x})\right]_+ \qquad h_j^{1=1} = \sum_i^{n^{l=0}} w_{ji}^{l=0} x_i \tag{2}$$

An output neuron is a linear readout, defined as $h_j^{L+1}$, where $L$ is the number of hidden layers. The bias terms in all layers are designed to be dependent on the activities of the previous layer, such that only a fixed fraction $f$ of the neurons in layer $l$ is active. This is similar to the adaptive bias used in sparse Top-k Transformer models [9, 10].

We investigate the network in the infinite-width regime, where we take the number of neurons in the previous layer $(n^{l-1})$ to infinity. There are a couple of benefits of studying this limit: (1) the posterior (Bayes optimal) $w$ becomes data-independent, (2) Due to the central limit theorem (CLT), it is tractable to compute the similarity, i.e. kernel, of neural representations of a pair of inputs. The kernel can be used to train the model.

From here on, we investigate a statistics of a single neuron in an intermediate layer, so we drop the subscript $j$ from the notations presented in Eqn. 1 for brevity. At the infinite width limit, $h^l$ is always a Gaussian distribution, if $w_i$ and $x_i^{l-1}$ are independent across all $i$'s, regardless of the shape of the distributions. Therefore, $x^l$ is a rectified Gaussian distribution. Note that the bias $b(\mathbf{x}^{l-1})$ is a deterministic positive value that shifts the mean of $h^l$ to a negative value (Eqn. 1). In order to make $h^l$ to be positive with probability $f$, the bias should be

$$b(\mathbf{x}^{l-1}) = \sigma_h(\mathbf{x}^{l-1})\tau_f \qquad \tau_f = \sqrt{2}\mathrm{erf}^{-1}(1 - 2f) \tag{3}$$

where $\sigma_h(\mathbf{x}^{l-1})$ is the standard deviation of $h^l$ which is dependent on $\mathbf{x}^{l-1}$. With a slight abuse of notation, $b(\mathbf{x}^{l-1})$ will be denoted just $b^l$ and $\sigma_h(\mathbf{x}^{l-1})$ as $\sigma_h$ with the dependence on $x^l$ implied. $\mathrm{erf}^{-1}(\cdot)$ is the inverse error function.

The similarity between neural representations (post-activations) at layer $l$ of two inputs $p$ and $q$ averaged over the random weights is known as the NNGP kernel $K^l(\mathbf{x}^{(p)}, \mathbf{x}^{(q)})$, where $\mathbf{x}^{(p)}$ is a vector notation of the input sample $p$. The formula of the kernel is given by

$$K^l(\mathbf{x}^{(p)}, \mathbf{x}^{(q)}) := \left\langle \left[ h^{l,(p)} - b^l \right]_+ \left[ h^{l,(q)} - b^l \right]_+ \right\rangle_{P(h^{l,(p)}, h^{l,(q)})} \tag{4}$$

where $h^{l,(p)}$ is the pre-activation at layer $l$ for input sample $p$. $\langle \cdot \rangle_{P(h^{l,(p)}, h^{l,(q)})}$ is the average over the joint distribution over $h^{l,(p)}$ and $h^{l,(q)}$. Analogous to the NNGP (post-activation) kernel, a pre-activation kernel $K_h^l$ computes the similarity of pre-activations of two inputs: $K_h^l(\mathbf{x}^{(p)}, \mathbf{x}^{(q)}) = \mathrm{Cov}\left[ h^{l,(p)}, h^{l,(q)} \right]$. For the sparse NNGP kernel, the relationship between $K^l$ and and $K_h^l$ is given by [12]:

$$K^l(\mathbf{x}^{(p)}, \mathbf{x}^{(q)}) = \frac{1}{2\pi}\sqrt{K_h^l(\mathbf{x}^{(p)}, \mathbf{x}^{(p)})K_h^l(\mathbf{x}^{(q)}, \mathbf{x}^{(q)})}\, g\left(\theta^l, f\right) \tag{5}$$

$$g\left(\theta^l, f\right) = 2I\left(\theta^l \mid \tau_f\right) - \tau_f\sqrt{2\pi}(1 + \cos\theta^l) \tag{6}$$

$$I(\theta \mid \tau_f) = \int_0^{\frac{\pi-\theta}{2}} \exp\left(-\frac{\tau_f^2}{2\sin^2(\phi_0)}\right) 2\sin\left(\phi_0 + \theta\right)\sin(\phi_0)$$

$$+ \tau_f\left(\sin\left(\phi_0 + \theta\right) + \sin(\phi_0)\right)\sqrt{\frac{\pi}{2}}\mathrm{erf}\left(\frac{\tau_f}{\sqrt{2}\sin(\phi_0)}\right)d\phi_0 \tag{7}$$

$I(\theta \mid \tau_f)$ can be efficiently computed with numerical integration. In Eqn.5, $\sqrt{K_h^l(\mathbf{x}^{(p)}, \mathbf{x}^{(p)})}$ is geometrically analogous to a length of the pre-activation for input $p$ at layer $l$. Typically, this is not an interesting part of a kernel, since the lengths are the same for length-normalized inputs. The part that gives the rich characteristics of the kernel is a complicated function $g(\cdot, \cdot)$, which takes the activation level $f$, and the angle $\theta^l$ between the two pre-activation representations at layer $l$, and returns a scalar. Formally, the angle is defined as

$$\theta^l = \arccos\frac{K_h^l(\mathbf{x}^{(p)}, \mathbf{x}^{(q)})}{\sqrt{K_h^l(\mathbf{x}^{(p)}, \mathbf{x}^{(p)})K_h^l(\mathbf{x}^{(q)}, \mathbf{x}^{(q)})}} \tag{8}$$

The pre-activation kernel is, in turn, dependent on the post-activation kernel of the previous layer.

$$K_h^l(\mathbf{x}^{(p)}, \mathbf{x}^{(q)}) = \sigma^2 K^{l-1}(\mathbf{x}^{(p)}, \mathbf{x}^{(q)}) \tag{9}$$

where $\sigma$ is the standard deviation of the weights. For the input layer, $l = 0$, the post-activation kernel $K^{l=0}$ is simply a dot product between a pair of input vectors. To train the NNGP, we use the kernel regression without the ridge factor. This is equivalent to training only the last layer, which is optimal in the Bayesian neural network formulation at the infinite width limit.

## 3. Weight correlations

In sparse networks, the neural representations of a pair of inputs rapidly become more and more dissimilar, i.e. $\theta^l$ increases, over layers, which loses the information of the inputs [12]. To counter this effect, we introduce a weak correlation between the random weights that still allow the CLT.

### 3.1. Central limit theorem for $m$-dependent stationary sequences

Here we review the CLT for $m$-dependent stationary sequence [16]. Consider a set $\{a_1, \ldots, a_i, \ldots, a_n\}$ of random variables with mean $0$ and variance $\sigma_a^2$. The set is called an $m$-dependent stationary sequence when any $a_i$ and $a_j$ are independent if and only if $|i - j| > m$. Define a sum $h = a_1 + \cdots + a_n$. The CLT for $m$-dependent stationary sequences states that the distribution of $h$ converges to a normal distribution with $0$ mean and standard deviation $\sigma_a \sqrt{n}$ as $n \to \infty$. This illustrates that, despite the sequence having dependencies up to $m$ terms apart, the normalized sum $h$ will still approach a Gaussian distribution. It is generally assumed that $m$ is a finite integer. In the context of our neural network, $a_i = w_i x_i$ is the summand for computing the pre-activation $h$ (dropping the superscripts for brevity). To create an $m$-dependent sequence of $a_i$, we require that the weights $w_i$ are $m$-dependent (Figure 1a).

### 3.2. Convolutional layer interpretation

Here we highlight that having $m$-dependent weights is equivalent to having an intermediate convolutional layer. Consider a weight matrix $\mathbf{W}$ whose each row is composed of an $m$-dependent sequence of random weights. The pre-activation vector in layer $l + 1$ is then given by $\mathbf{h}^{l+1} = \mathbf{W} \mathbf{x}^l$. Such a matrix $\mathbf{W}$ can be factorized into i.i.d. random weights matrix $\mathbf{U}$ and a banded Toeplitz matrix $\mathbf{C}$ with value 1's on the band: $\mathbf{W} = \mathbf{U} \mathbf{C}$. The linear operation by $\mathbf{W}$ on $\mathbf{x}^l$ is therefore equivalent to

$$\mathbf{h}^{l+1} = \mathbf{U} \mathbf{x}^{l+0.5} \qquad \mathbf{x}^{l+0.5} = \mathbf{C} \mathbf{x}^l \tag{10}$$

The Toeplitz matrix $\mathbf{C}$ carries out the linear convolution over neurons in layer $l$. The convolutional filter locally sums over $m + 1$ neurons. We thereby show that having $m$-dependent weights is equivalent to having a linear intermediate convolutional layer $l + 0.5$ between the layer $l$ and $l + 1$, followed by a fully connected layer with i.i.d. random weights (Figure 1b). The convolutional operation is analogous to the local lateral connection to neighboring neurons in the cortex, whereas the random i.i.d. weights are analogous to the long-range projections in the cortex.

### 3.3. Kernel computation

Computing the post-activation kernel $K^l$ from the pre-activation kernel $K_h^l$ remains the same as the independent weights case (Eqn. 5). The only difference is in computing the pre-activation kernel $K_h^l$ from the post-activation kernel of the previous layer $K^{l-1}$ (for the independent case we had Eqn. 9). The more general form of Eqn. 9 is given by

$$K_h^l(\mathbf{x}^{(p)}, \mathbf{x}^{(q)}) = \sigma^2 K^{l-1}(\mathbf{x}^{(p)}, \mathbf{x}^{(q)}) + \sigma^2 m \frac{1}{2\pi} \sqrt{K_h^{l-1}(\mathbf{x}^{(p)}, \mathbf{x}^{(p)}) K_h^{l-1}(\mathbf{x}^{(q)}, \mathbf{x}^{(q)})} g\left(\theta = \frac{\pi}{2}, f\right) \tag{11}$$

The derivation is presented in Appendix B. Equivalently, substituting $K^{l-1}$ using Eqn. 5, we have

$$K_h^l(\mathbf{x}^{(p)}, \mathbf{x}^{(q)}) = \frac{\sigma^2}{2\pi} \sqrt{K_h^{l-1}(\mathbf{x}^{(p)}, \mathbf{x}^{(p)}) K_h^{l-1}(\mathbf{x}^{(q)}, \mathbf{x}^{(q)})} \left[ g\left(\theta^{l-1}, f\right) + m g\left(\frac{\pi}{2}, f\right) \right] \tag{12}$$

When the weights are independent ($m = 0$), Eqn. 11 reverts to Eqn. 9 by eliminating the second term. Note that the value of $g\left(\frac{\pi}{2}, f\right)$ in the second term is a constant for a given $f$. Therefore, if we assume $\sqrt{K_h^{l-1}(\mathbf{x}^{(p)}, \mathbf{x}^{(p)})}$ is the same for all inputs $p$'s, which is the case for normalized inputs, the second term of Eqn. 11 is input-independent but may change over layers. This is similar to having a noisy bias in the pre-activation, although the bias noise would stay constant across the layers. For the noisy bias case, the second term in Eqn. 11 would be replaced with a variance of the bias noise.

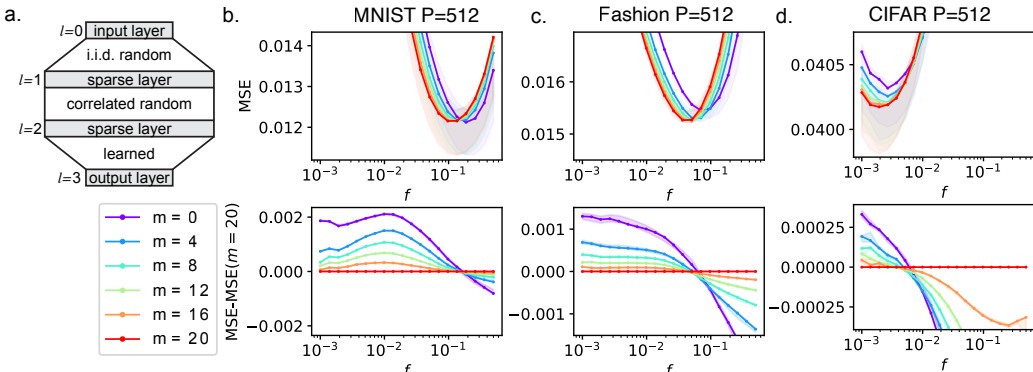

Figure 2: Experimental observations of the simplest network model with sparse activation and correlated weights. (a) 2-hidden layer neural network diagram. 50% confidence intervals across 8 trials of random training sets are shown with shades. (b) Results on MNIST (training set size: $P = 512$, test set size: $2P$). Top: The median (over 8 trials) generalization performance of the model with varying degrees of activation levels $f$ and correlation levels $m$. The mean squared error (MSE) on the test set is reported. Bottom: To compare the performance curves, the performance curve of the highest weight correlation $m = 20$ is subtracted from the rest of the curves. Where the difference is positive, $m = 20$ outperformed, and where negative, $m = 20$ underperformed. Generally, $m = 20$ outperforms in a sparser regime. Solid lines: median over 8 trials. 50% confidence intervals are shown with shades. (c) Results on Fashion-MNIST dataset. (d) Results on CIFAR10. More are shown in Appendix E

.

Note that we need a deterministic bias in order to induce sparsity, so inducing the noisy bias-like effect using the correlated weights is a viable alternative to actually having a noisy bias. In a kernel formulation for the more general form of correlated weights, the effect of correlation becomes a lot more complicated than that of the noisy bias (see Appendix B).

## 4. Numerical experiments

The simplest network to investigate the effect of correlated weights is the network with 2 hidden layers, with the correlated weights between the first and the second hidden layers (Fig. 2a). We do not want the correlated weights between the input and the first hidden layer, since having them is equivalent to simply replacing the input vectors with blurred (i.e. convolved with a convolutional kernel with value 1's) versions of the input vectors. Also, the aim of this paper is to investigate how the correlated weights counter the random sparse activities in the presynaptic neurons. As for the readout weights, we do not consider the weight correlation, since the readout weights will be trained without the ridge factor (i.e. $L_2$ regularizer), thereby ignoring the prior over the readout weights.

We trained the 2-hidden layer model with varying degrees of weight correlations $m$ and activation levels $f$ on the MNIST, Fashion-MINST, and CIFAR10 datasets [17–19] (Figure 2b-d). The training is performed by regressing on the 10-dimensional one-hot vectors using kernel ridgeless regression. Although the datasets were meant for the classification, it has been widely used for benchmarking regression performances of NNGP models [15, 20]. We consistently observe that higher weight correlation, i.e. larger $m$, improves generalization performance in the sparser regime, but degrades the performance in the denser regime. This observation is consistent for all tested datasets of varying numbers of training set sizes $P$, with very little variation across 8 trials of randomly selected training samples (see Appendix E for comprehensive results). Across the paper, the generalization performances were measured on test sets twice the sizes of the training sets. In the case of Fashion-MINST and CIFAR10, the sparse models with correlated weights have the best performances.

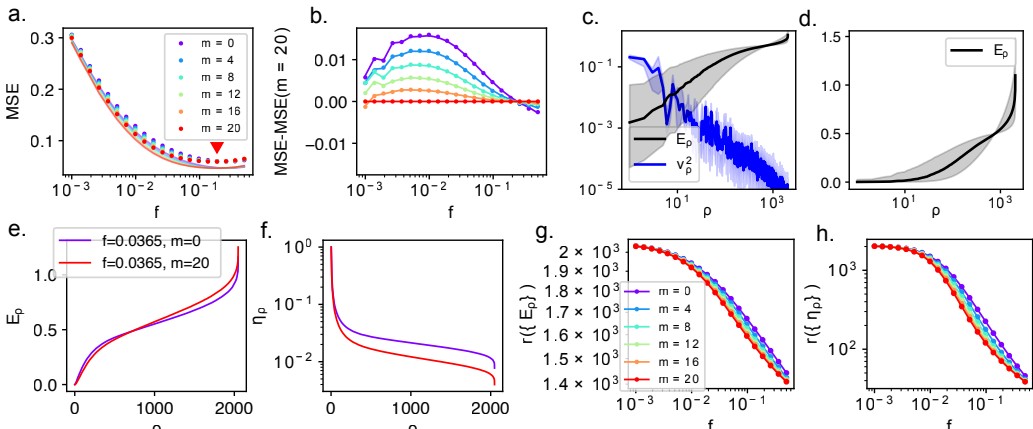

Figure 3: Generalization theory of the 2-hidden layer NNGP models with correlated weights and sparse activation. (a) Dotted markers: empirically obtained generalization error on two-class (digits 3 and 5) MNIST dataset. Solid lines: theoretically obtained generalization errors. (b) The performance curve of the $m = 20$ case subtracted from the other curves. Positive: $m = 20$ outperforms. Negative: $m = 20$ underperforms. Dotted: empirical. Solid line: theory. (c) Blue: the median target function spectra $v_\rho^2$ for different model configurations. Black: the median modal error spectra $E_\rho$ for different model configurations. Shades: 50% confidence intervals. (d) Same as (c) but on different axes scales. (e) Modal error spectra for two different correlation levels. (f) Same as (e) but for eigen spectra. (g) Participation ratios of modal error spectra for all models. (h) Same as (g) but for eigen spectra.

## 5. Generalization performance analysis

The theoretical explanation of the experimental observation is twofold. First, using the results by Canatar et al. [15], we theoretically review that kernel Gram matrix with moderately low dimensionality has a high generalization performance, and empirically show that the weight correlation yields a low-dimensional kernel Gram matrix. This is in contrast to the observation that sparsity yields a high-dimensional Gram matrix [12]. Second, we establish a theoretical explanation of why the weight correlation yields a low-dimensional kernel Gram matrix. In conjunction, these two parts presented in the next two sections (5.1,5.2) complete the theoretical explanation of the connection between the weight correlation and improvement in the generalization performance in the sparse regime. In the final third section 5.3, we derive an optimal weight correlation level $m$ (for a given activation level $f$) that balances the dimensionality-increasing force of the sparsity and the dimensionality-decreasing force of the weight correlation. We then show that the best generalization performance is attained at this balance in deep neural networks.

### 5.1. Moderately low-dimensional kernel Gram matrix improves the generalization performance

We start by reviewing the theory of the generalization performance of kernel regression by Canatar et al. [15]. Let $\eta_\rho$ and $\phi_\rho(\cdot)$ be the $\rho^{th}$ eigenvalue and the eigenfunctions of a kernel given by a Mercer decomposition. If the target function $\bar{f} : \mathbb{R}^{n_0} \to \mathbb{R}$ lives in a reproducing kernel Hilbert space (RKHS) given by the kernel, then the target function can be expressed in terms of the coordinates $(v_\rho)$ on the eigenfunctions: $\bar{f}(\mathbf{x}) = \sum_{\rho=0}^{N-1} v_\rho \phi_\rho(\mathbf{x})$ where $N$ is the number of non-zero eigenvalues. At the large training sample size and large $N$ limit, the expected generalization error $E_g$ computed via replica trick is given as a sum of modal errors $E_\rho$ weighted by the target powers $v_\rho^2$.

$$E_g = \sum_\rho v_\rho^2 E_\rho \tag{13}$$

$$E_\rho = \frac{1}{1-\gamma} \frac{\kappa^2}{(\kappa + P\eta_\rho)^2} \qquad \gamma = \sum_\rho \frac{P\eta_\rho^2}{(\kappa + P\eta_\rho)^2} \qquad \kappa = \sum_\rho \frac{\kappa\eta_\rho}{\kappa + P\eta_\rho} \tag{14}$$

where $P$ is the size of the training set. Note that $E_\rho$ is independent of the target function but dependent on the kernel which is encapsulated by the eigenspectrum (set of $\eta_\rho$). Also, the target function spectrum (set of $v_\rho^2$) is not only dependent on the target function itself, but also on the kernel. Therefore, when comparing the generalization performances of kernels for the same target function, one needs to, in principle, compare both the modal error spectra (set of $E_\rho$) and target function spectra.

We observe that the theory predicts the generalization performances of NNGP kernels with varying degrees of sparsity and weight correlation (Figure 3ab). As an illustrative example, we used the MNIST dataset with two classes (digits 3 and 5) and the target outputs are scalar values $-1$ and $1$ for labeling the two digits.

Agreeing with the observation made in Chun and Lee [12], the target function spectrum does not differ much across different kernels, compared to the large variation in the modal error spectrum (compare the confidence intervals of modal error and target function spectra in Figure 3cd). Therefore, we only need to compare the change in the modal error spectrum to explain the difference in the generalization performance between the kernels. It is shown in Figure 3e that for a fixed sparsity value $f = 0.0365$, a higher weight correlation leads to a steeper modal error spectrum. At this sparsity level, the higher weight correlation improves the performance. Since the generalization error $E_g$ is the sum of $E_\rho$ weighted by the target function powers $v_\rho^2$, it is beneficial to have a small value of $E_\rho$ at $\rho$'s where the weights for the sum, i.e. $v_\rho^2$, is large. This is achieved by having a steep modal error spectrum. However, when the modal error spectrum is too steep, the $E_\rho$ at the rest of the $\rho$ may become too large, contributing to an increase in the generalization error.

We observe that a steeper modal error spectrum is attained when the eigenspectrum is steep, i.e. the kernel has low effective dimensionality (Figure 3f). This relationship is theoretically supported in Chun and Lee [12]. To quantify the change in the steepness of the modal error spectra and the eigenspectra, we compute the participation ratios of the spectra, given by $r(\{a_i\}) = (\sum_i a_i)^2/(\sum_i a_i^2)$, where $\{a_i\}$ denotes a set whose elements are some values $a_i$ for some $i$'s. A small participation ratio indicates a steeper spectrum. Figure 3gh shows that the higher weight-correlations have smaller participation ratios for both modal error- and eigen-spectra (the red curve is the lowest). When the fraction of active neurons is large (large $f$; low sparsity), the participation ratio is very low, indicating a very steep modal error spectrum. As noted earlier, an excessively steep modal error spectrum can increase $E_g$, which explains the increase in $E_g$ at the dense regime (see $f > 0.2$ in Figure 3a). On the other hand, in the sparser regime where the eigenspectra are moderately steep, i.e. moderately low-dimensional, the higher weight correlation further lowers the dimensionality, improving the generalization performance, assuming the correlation is not excessive.

## 5.2. Weight correlation yields low-dimensional kernel Gram matrix

Thus far, we have observed that a higher weight correlation lowers the rank of the kernel Gram matrix, which helps with improving the generalization performance in the sparse regime. In this section, we explain how the higher weight correlation lowers the rank of the kernel Gram matrix. To this end, we return our attention to the recursive formula for computing the pre-activation kernel Eqn. 12. Note that for understanding the dimensionality of the kernel over layers, one can gain practically the same insight from studying either the post-activation kernel or the pre-activation kernel, since these two are related by a monotonically increasing function $g(\cdot, f)$.

We can ignore the change in the length of pre-activation, i.e. $\sqrt{K_h^{l-1}(\mathbf{x}^{(p)}, \mathbf{x}^{(p)})}$, over layers, by choosing a standard deviation of the weights that maintains the length constant in all layers [21, 22]. We denote this critical standard deviation as $\sigma^*$ (see Appendix C for derivation). With $\sigma^*$, the length of pre-activation at all layers is $\sigma^*\|\mathbf{x}\|$ where $\|\mathbf{x}\|$ is the length of an input vector. In this case, we can divide both sides of Eqn. 12 by $\sqrt{K_h(\mathbf{x}^{(p)}, \mathbf{x}^{(p)})K_h(\mathbf{x}^{(q)}, \mathbf{x}^{(q)})}$ and obtain the following dynamical

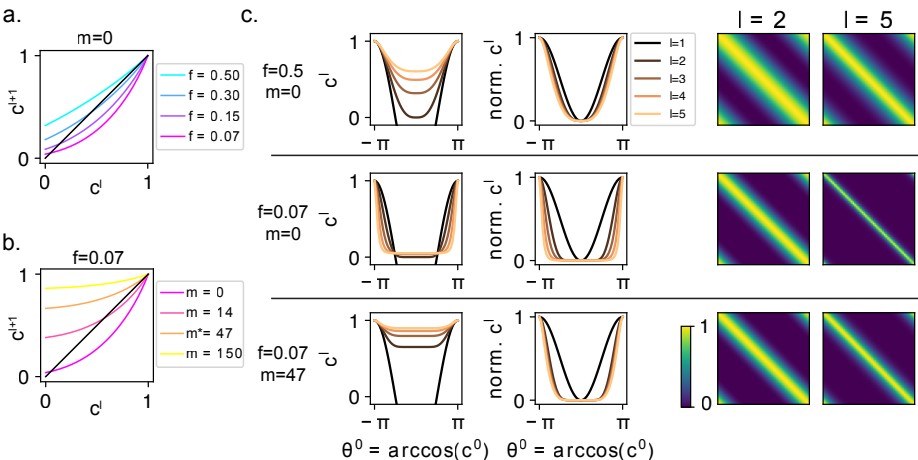

Figure 4: Dynamics of the NNGP kernels with sparsity and correlated weights. (a) The relationship between $c^l$ and $c^{l+1}$ for different levels of $f$ for $m = 0$. The intersection between the line of unity (black) and the curves are the fixed points. (b) Same as (a) but with different levels of $m$ for $f = 0.07$. (c) Top row: the dynamics of $c^l$ over layers. The $1^{st}$ (left-most) column: the raw $c^l$ values. the x-axis is the angle between the input vectors. The $2^{nd}$ column: the shift-normalized $c^l$ values. The $3^{rd}$ and $4^{th}$ columns: example Gram matrices at different layers. Second row and third row: different model configurations.

equation:

$$c^l = \frac{\sigma^{*2}}{2\pi} \left[ g\left(\arccos c^{l-1}, f\right) + mg\left(\frac{\pi}{2}, f\right) \right] \tag{15}$$

where $c^l$ is the pre-activation correlation of two inputs: $c^l := \frac{K_h^l(\mathbf{x}^{(p)}, \mathbf{x}^{(q)})}{\sqrt{K_h(\mathbf{x}^{(p)}, \mathbf{x}^{(p)}) K_h(\mathbf{x}^{(q)}, \mathbf{x}^{(q)})}} = \cos \theta^l$. If the input vectors lie on a unit ball, $c^l$ is just a scaled version of $K_h^l$. The iterative equation of $c^l$ is visualized in Figure 4ab for different values of $f$ and $m$. Note that the fixed points of the iterative equation where $c^l = c^{l-1}$ are visualized in Figure 4ab by the intersections of the function and the line of unity. It is immediately noticeable that one of the fixed points is always at $c^l = 1$ (Figure 4ab), which corresponds to $\theta^l = 0$. As a trivial example, this means that $c^l$ between two identical inputs, i.e. normalized length of the input, stays constant throughout all layers.

It is more interesting to study the stability of the fixed point $c^l = 1$. When the fixed point is unstable (the slope at $c^l = 1$ is greater than 1), such as in the cases of $f < 0.5$ $m = 0$ in Figure 4a, a pair of inputs vectors with $c^{l=0} < 1$ will have $c^l$ value decrease away from 1 towards a stable fixed point $c^l < 1$. This means that any pair of neural representations dissimilates over layers for such kernels. These kernels are referred to as being in a disordered regime [20–22]. For kernels with i.i.d. weights ($m = 0$), only the dense $f = 0.5$ kernel has slope 1 at $c^l = 1$ and all sparser $f < 0.5$ kernels are in the disordered regime (Figure 4a). However, with an introduction of the correlated weights, one can bring the sparse model into the ordered regime where the slope is less than 1 at $c^l = 1$. Figure 4b shows an example case where a sparse kernel with $f = 0.07$ moves from the disordered regime to the ordered regime with a large weight correlation. In summary, the sparsity always dissimilates the neural representations whereas the weight correlation always assimilates the neural representations. This means that weight correlation makes the kernel Gram matrix more uniform over the layers, making the Gram matrix have low dimensionality.

## 5.3. Optimal weight correlation

Here we present the formula for optimal weight correlation as a function of sparsity $f$. This is based on the observations that the kernels along the phase boundary of the disordered and ordered regimes have the best performance in deep NNGP kernels [20]. The optimal weight correlation $m^*$

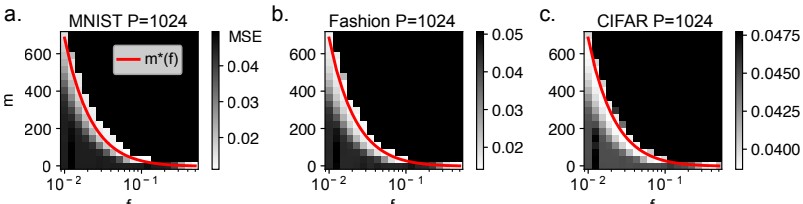

Figure 5: Empirical generalization performances of the deeper networks (L=17) on (a) MNIST, (b) Fashion-MNIST and (c) CIFAR10 datasets. Grayscale values report the MSEs on the test dataset. Red line is the phase boundary that determines the theoretical optimal weight correlation degree $m^*$ as the function sparsity $f$. Above the boundary is the ordered regime, and below the boundary is the disordered regime.

that puts a kernel with sparsity $f$ in the phase boundary ($\frac{dc^{l+1}}{dc^l} = 1$) is given by

$$m^*(f) = \frac{\frac{d}{dc}g(\arccos c, f)\big|_{c=1} - g(0, f)}{g\left(\frac{\pi}{2}, f\right)} \tag{16}$$

Computing the derivative is non-trivial (see Appendix D for the full formula).

The values of $c^l$ as the function of the angle $\theta^0$ between a pair of input vectors are shown in the first column of Figure 4c. The kernel with $f = 0.5$ and $m^* = 0$ (Figure 4c top) is in the boundary of the disordered and ordered regimes as stated earlier. As expected, $c^l$ values converge to $1$ albeit slowly. The same is true for the kernel with $f = 0.07$ and $m^* = 47$ (Figure 4c bottom) which is also in the boundary. However, the kernel in the disordered regime (Figure 4c middle) has $c^l$ values converging to $c^l \ll 1$. It is not immediately obvious that the boundary case kernels are said to have optimal performances when the kernel values assimilate.

With $\sigma^*$ and the unit ball input assumption, $c^l$ is a scaled version of $K_h^l$ and a scaled-shifted version of $K^{l-1}$. It has been empirically and theoretically shown that the scale and shift of $K^{l-1}$ typically do not affect the generalization performance, and it is the shape of $K^{l-1}$ that matters [12, 15]. Therefore, we can normalize the $c^l$ by scaling and shifting such that the normalized $c^l$ always ranges from $0$ to $1$. This helps us to visualize the change in the shape of $c^l$ that matters to the generalization performance (Figure 4c $2^{nd}$ column). The second column of Figure 4c shows that the boundary case kernels (top, bottom) maintain the shape of $c^l$ over layers, without much difference between layers. On the other hand, the kernel in the disordered regime (middle) loses its shape rapidly over layers, quickly converging to the shape of the Dirac delta function. For the kernel in the ordered regime, the $c^l$ values converge too quickly to $1$, bringing numerical instability when training at deeper layers. Therefore, only the kernels on the boundary can maintain high performance in a deeper layers.

Through numerical experiments, we confirm our theory (Figure 5). A deep NNGP with 17 hidden layers ($L = 17$) has the best performance at the phase boundary $m^*(f)$ given by Eqn. 16 (red line in Figure 5). The results from NNGP's of different depths are shown in Appendix G.

## 6. Discussion

There is much discussion about why sparse networks are universally observed along with their computational benefits. In this work, we introduce a mechanism for correlated weights and study its effect on the performance of infinitely-wide sparse neural networks. We show that both theoretically and empirically, the correlated weights increase generalization performance in sparse random neural networks. We also show how to compute the optimal correlation level that gives the best-performing kernel in deep and sparse random networks. To this end, we utilize the recent theoretical results on kernel regression [12, 15] to explain why these networks perform the best across the range of hyperparameters. Future work will investigate more sophisticated forms of correlations and compare their performance across a range of neural network architectures.

## Acknowledgements

We would like to thank Haim Sompolinsky for an insightful early discussion on NNGP, and Lawrence Saul for valuable feedback on a draft of the manuscript. Research reported in this publication was supported by the National Institutes of Health under award number U19NS104653.

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

## A. Code Availability

The codes for our computational methods and figure generation are available at: `https://github.com/badooki/Sparse_and_Correlated/`.

## B. Generalized correlated weights kernel

Consider a random correlated matrix $\mathbf{W}$ constructed by convolving independent random matrix $\mathbf{U}$ with a constant filter matrix $\mathbf{V}$ of size $z_h \times z_w$ where $z_h$ and $z_w$ are odd numbers. The resulting element of $\mathbf{W}$ is given as

$$w_{i,j} = \sum_{\beta=1}^{z_w} \sum_{\alpha=1}^{z_h} \mathbf{U}_{i-\frac{z_h+1}{2}+\alpha, j-\frac{z_w+1}{2}+\beta} \mathbf{V}_{\alpha,\beta}$$

The covariance between two arbitrary elements $w_{ij}$ and $w_{kl}$ is given by

$$\text{Cov}\left[w_{i,j}, w_{k,l}\right]$$
$$= \begin{cases} \text{Var}\left[u\right] \sum_{\beta=1}^{z_w-|l-j|} \sum_{\alpha=1}^{z_h-|k-i|} \mathbf{V}_{\alpha,\beta} \mathbf{V}_{|k-i|+\alpha, |l-j|+\beta} & (|k-i| < z_h) \wedge (|l-j| < z_w) \\ 0 & \text{otherwise} \end{cases} \quad (17)$$

where $\text{Var}\left[u\right]$ is the variance of the elements of $\mathbf{U}$.

A special case of this formulation is where $\mathbf{V}$ is of rank 1. If $\mathbf{V} = \mathbf{p}\mathbf{q}^\top$,

$$\mathbf{W} = \mathbf{P}\mathbf{U}\mathbf{Q}$$

where $\mathbf{P}$ and $\mathbf{Q}$ are band matrices with bandwidths $\frac{z_h-1}{2}$ and $\frac{z_w-1}{2}$ respectively where the non-zero elements are centered at the diagonals of the matrices. For each row of $\mathbf{P}$, the non-zero elements are given by $\mathbf{p}^\top$ with their orders preserved, and the same for $\mathbf{Q}$. As a linear operator, $\mathbf{W}$ takes input $\mathbf{x}$, convolve it with $\mathbf{q}$, multiplies the resulting vector with a random matrix $\mathbf{U}$, and finally convolves the resulting vector with $\mathbf{p}$.

If $\mathbf{V}$ is a matrix with all elements value 1, that corresponds to the rank-1 case with $\mathbf{p} = \mathbf{1}$ and $\mathbf{q} = \mathbf{1}$. In this case, the formula for the covariance is simplified to

$$\text{Cov}\left[w_{i,j}, w_{k,l}\right] = \text{Var}\left[u\right] \left(z_h - \min\left(|k-i|, z_h\right)\right) \left(z_w - \min\left(|l-j|, z_w\right)\right)$$

To maintain $Var[w_{ij}] = \frac{\sigma^2}{N}$, we need to choose the variance of $u$ to be $\frac{\sigma^2}{z_h z_w N}$, where $N$ is the number of neurons in the previous layer. The rescaled covariance is

$$\text{Cov}\left[w_{i,j}, w_{k,l}\right] = \frac{\sigma^2}{z_h z_w N} \left(z_h - \min\left(|k - i|, z_h\right)\right)\left(z_w - \min\left(|l - j|, z_w\right)\right)$$

The weight matrix $\mathbf{W}$ has $m$-dependent sequence of weights in both row and column, where $m \equiv z_w - 1 = z_h - 1$. We call it a doubly $m$-dependent matrix.

## B.1. Correlated neurons as the result of the correlated weights

Here we show that the doubly $m$-dependent zero-mean weight matrix results in $m$-dependent output neurons regardless of the statistics of the input.

$$h_i = \sum_j w_{ij} x_j$$

$$\text{Cov}[h_i, h_k] = \sum_j \text{Cov}[w_{ij}x_j, w_{kj}x_j] + \sum_{j \neq l} \text{Cov}[w_{ij}x_j, w_{kl}x_l] \tag{18}$$

$$= \sum_j E[w_{ij}x_j w_{kj}x_j] - E[w_{ij}x_j]E[w_{kj}x_j] + \sum_{j \neq l} E[w_{ij}x_j w_{kl}x_l] - E[w_{ij}x_j]E[w_{kl}x_l] \tag{19}$$

Assuming that $w_{ij}$ and $x_j$ are independent from each other and $E[w_{ij}] = 0$,

$$\text{Cov}[h_i, h_k] = \sum_j \text{Cov}[w_{ij}w_{kj}]E[x_j^2] + \sum_{j \neq l} \text{Cov}[w_{ij}w_{kl}]E[x_j x_l]$$

Acknowledging the fact that $\mathbf{W}$ is doubly $m$-dependent, and assuming $x$ is a stationaty $m$-dependent sequence,

$$\text{Cov}[h_i, h_k] = N\text{Cov}[w_{ij}w_{kj}]E[x^2] + 2N\sum_{l=1}^{m} \text{Cov}[w_{i0}w_{kl}]E[x_0 x_l]$$

It is evident that $h$ is a stationary $m$-dependent sequence.

## B.2. Summary on the pre-activation correlations

For deep neural network, the post-activation of cell $j$ in layer $l$ for stimulus $p$ is denoted $x_j^{(p)l}$, and the pre-activation is denoted $h_j^{(p)l}$. The model is therefore written as

$$x_j^{(p)l} = \phi(h_j^{(p)l} - b)$$

$$h_j^{(p)l} = \sum_i w_{ji} \cdot x_i^{(p)l-1}$$

The within-cell correlation is

$$\text{Cov}[h^{(p)l}, h^{(q)l}] = \sigma^2 E[x^{(p)l-1}x^{(q)l-1}] + 2N\sum_{g=1}^{m} \text{Cov}[w_{i0}, w_{ig}]E[x_0^{(p)l-1}x_g^{(q)l-1}]$$

The inter-cell correlation is

$$\text{Cov}[h_i^{(p)l}, h_k^{(q)l}] = N\text{Cov}[w_{ij}, w_{kj}]E[x^{(p)l-1}x^{(q)l-1}] + 2N\sum_{g=1}^{m} \text{Cov}[w_{i0}, w_{kg}]E[x_0^{(p)l-1}x_g^{(q)l-1}]$$

where

$$\text{Cov}[w_{ij}, w_{kl}] = \frac{\sigma^2}{z_h z_w N} \left(z_h - \min\left(|i - k|, z_h\right)\right)\left(z_w - \min\left(|j - l|, z_w\right)\right)$$

and $z = m + 1$. In summary, after some clean-up, the above is written as the following.

The within-cell pre-activation correlation is

$$\text{Cov}[h^{(p)l}, h^{(q)l}] = \sigma^2 E[x^{(p)l-1} x^{(q)l-1}] + 2 \frac{\sigma^2}{z_h z_w} \sum_{g=1}^{m} f_{i0ig} E[x_0^{(p)l-1} x_g^{(q)l-1}]$$

The inter-cell pre-activation correlation is

$$\text{Cov}[h_i^{(p)l}, h_k^{(q)l}] = \frac{\sigma^2}{z_h z_w} f_{ijkj} E[x^{(p)l-1} x^{(q)l-1}] + 2 \frac{\sigma^2}{z_h z_w} \sum_{g=1}^{m} f_{i0kg} E[x_0^{(p)l-1} x_g^{(q)l-1}]$$

where

$$f_{ijkl} = (z_h - \min(|i - k|, z_h))(z_w - \min(|j - l|, z_w))$$

$$z = m + 1$$

## B.3. Post-activation correlations

Let $\mathbf{x}^{(p)}$ be the input vector for a sample $p$, and $x^{(p)l}$ be the activation in layer $l$ for the same sample. The within-cell post-activation kernel $K^l(\mathbf{x}^{(p)}, \mathbf{x}^{(q)}) = E[x^{(p)l} x^{(q)l}]$ and inter-cell post-activation kernel $K_{ik}^l(\mathbf{x}^{(p)}, \mathbf{x}^{(q)}) = E[x_i^{(p)l} x_k^{(q)l}]$ in layer $l$ can be written in terms of the pre-activation kernels in layer $l$, for example, by using the sparse NNGP formula.

The within-cell post-activation kernel is

$$K^l(\mathbf{x}^{(p)}, \mathbf{x}^{(q)}) = \frac{1}{2\pi} \sqrt{K_h^l(\mathbf{x}^{(p)}, \mathbf{x}^{(p)}) K_h^l(\mathbf{x}^{(q)}, \mathbf{x}^{(q)})} \left( 2I \left( \theta^l \mid \tau \right) - \tau \sqrt{2\pi} (1 + \cos \theta^l) \right)$$

$$\theta^l = \arccos \frac{K_h^l(\mathbf{x}^{(p)}, \mathbf{x}^{(q)})}{\sqrt{K_h^l(\mathbf{x}^{(p)}, \mathbf{x}^{(p)}) K_h^l(\mathbf{x}^{(q)}, \mathbf{x}^{(q)})}}$$

where $I(\cdot|\cdot)$ is defined in the main text and $\tau$ is the same as $\tau_f$ defined in the main text. The inter-cell post-activation kernel is

$$K_{ik}^l(\mathbf{x}^{(p)}, \mathbf{x}^{(q)}) = \frac{1}{2\pi} \sqrt{K_h^l(\underline{x}^{(p)}, \underline{x}^{(p)}) K_h^l(\underline{x}^{(q)}, \underline{x}^{(q)})} \left( 2I \left( \theta_{(ik)}^l \mid \tau \right) - \tau \sqrt{2\pi} (1 + \cos \theta_{(ik)}^l) \right)$$

$$\theta_{(ik)}^l = \arccos \frac{K_{h(ik)}^l(\mathbf{x}^{(p)}, \mathbf{x}^{(q)})}{\sqrt{K_h^l(\mathbf{x}^{(p)}, \mathbf{x}^{(p)}) K_h^l(\mathbf{x}^{(q)}, \mathbf{x}^{(q)})}}$$

## B.4. From the input layer to the first hidden layer

In deep NNGP, we assume independent weights for $\mathbf{W}$ from the input layer to the first hidden layer. However, in case one wants to have dependent weights for that $\mathbf{W}$, the following shows how to compute the first hidden layer kernel.

$$K_{h(ik)}^{l=1}(\mathbf{x}^{(p)}, \mathbf{x}^{(q)}) = \sum_{j,l} \text{Cov}[w_{ij} x_j^{(p)}, w_{kl} x_l^{(q)}] \tag{20}$$

$$= \sum_{j,l} x_j^{(p)} x_l^{(q)} \text{Cov}[w_{ij}, w_{kl}] = \frac{\sigma^2}{z_h z_w N} \sum_{j,l} f_{ijkl} x_j^{(p)} x_l^{(q)} \tag{21}$$

## B.5. No correlation across post-synaptic neurons $z_h = 1$, $z_w > 1$

The weight correlation discussed in the main text is equivalent to the $z_h = 1$ case, and therefore $K_{h(ik)}^l(\mathbf{x}^{(p)}, \mathbf{x}^{(q)}) = 0$. Then $f_{i0ig} = z_w - \min(|0 - g|, z_w) = z_w - g$. This means we always have

$$\theta_{(ik)}^l = \arccos(0) = \frac{\pi}{2}$$

. Since the pre-activations of the post-synaptic neurons are independent $(K_{h(ik)}^l(\mathbf{x}^{(p)}, \mathbf{x}^{(q)}) = 0)$, the post-activations are also independent. This means that the inter-cell post-activation kernel is always the following for any pair of post-synaptic neurons.

$$K_{ik}^l(\mathbf{x}^{(p)}, \mathbf{x}^{(q)}) = \frac{1}{2\pi}\sqrt{K_h^l(\mathbf{x}^{(p)}, \mathbf{x}^{(p)})K_h^l(\mathbf{x}^{(q)}, \mathbf{x}^{(q)})}\left(2I\left(\frac{\pi}{2} \mid \tau\right) - \tau\sqrt{2\pi}(1 + \cos\frac{\pi}{2})\right)$$

We obtain the above by plugging $\theta_{(ik)}^l = \frac{\pi}{2}$ into the inter-cell post-activation kernel formula derived earlier.

Now derive the formula for the pre-activation kernel. Start with the within-cell pre-activation kernel formula derived earlier.

$$K_h^{l+1}(\mathbf{x}^{(p)}, \mathbf{x}^{(q)}) = \sigma^2 K^l(\mathbf{x}^{(p)}, \mathbf{x}^{(q)}) + 2\frac{\sigma^2}{z_h z_w}\sum_{g=1}^m f_{i0ig}K_{0g}^l(\mathbf{x}^{(p)}, \mathbf{x}^{(q)}) \tag{22}$$

$$= \sigma^2 K^l(\mathbf{x}^{(p)}, \mathbf{x}^{(q)}) + 2\frac{\sigma^2}{z_h z_w}K_{0g}^l(\mathbf{x}^{(p)}, \mathbf{x}^{(q)})\sum_{g=1}^m (z_w - g) \tag{23}$$

$$= \sigma^2 K^l(\mathbf{x}^{(p)}, \mathbf{x}^{(q)}) + \sigma^2 m K_{0g}^l(\mathbf{x}^{(p)}, \mathbf{x}^{(q)}) \tag{24}$$

By expanding the formula for $K^l(\mathbf{x}^{(p)}, \mathbf{x}^{(q)})$ and $K_{0g}^l(\mathbf{x}^{(p)}, \mathbf{x}^{(q)})$. Now we get a formula for the within-cell pre-activation kernel in terms of the within-cell pre-activation kernel of the previous layer.

$$K_h^{l+1}(\mathbf{x}^{(p)}, \mathbf{x}^{(q)}) = \sigma^2\frac{1}{2\pi}\sqrt{K_h^l(\mathbf{x}^{(p)}, \mathbf{x}^{(p)})K_h^l(\mathbf{x}^{(q)}, \mathbf{x}^{(q)})}$$
$$\times\left[\left(2I\left(\theta^l \mid \tau\right) - \tau\sqrt{2\pi}(1 + \cos\theta^l)\right) + m\left(2I\left(\frac{\pi}{2} \mid \tau\right) - \tau\sqrt{2\pi}\right)\right] \tag{25}$$

In fact, we do not need to compute the inter-cell kernels (for both pre-activation and post-activation).

Since

$$g\left(\theta^l, f\right) = 2I\left(\theta^l \mid \tau\right) - \tau\sqrt{2\pi}(1 + \cos\theta^l)$$

where $\tau = \sqrt{2}\mathrm{erf}^{-1}(1 - 2f)$, we get the expression shown in the main text

$$K_h^l(\mathbf{x}^{(p)}, \mathbf{x}^{(q)}) = \frac{\sigma^2}{2\pi}\sqrt{K_h^{l-1}(\mathbf{x}^{(p)}, \mathbf{x}^{(p)})K_h^{l-1}(\mathbf{x}^{(q)}, \mathbf{x}^{(q)})}\left[g\left(\theta^{l-1}, f\right) + mg\left(\frac{\pi}{2}, f\right)\right]$$

## B.6. No correlation across pre-synaptic neurons $z_w = 1$, $z_h > 1$

When $z_w = 1$, the independent randomness across the synaptic weights connected to one post-synaptic neuron and all pre-synaptic neurons makes the kernel computation identical to the regular independent weights NNGP kernel. The post-synaptic neurons become correlated due to $z_h > 1$, but this property does not have any effect on computing the activities of the next layer, since the independent weights across the row of $\mathbf{W}$ erases the correlated properties of the neurons. Therefore, the resulting NNGP kernels are identical to the i.i.d. random weights case.

## C. Deviation of $\sigma^*$

We want to find the weight variation $\sigma^{*2}$ that makes the neural lengths $\sqrt{K^l(\mathbf{x}^{(p)}, \mathbf{x}^{(p)})}$ to stay identical across layers. This is equivalent to requiring the pre-activation activation length to stay identical.

$$\frac{K_h^{l+1}(\mathbf{x}^{(p)}, \mathbf{x}^{(p)})}{K_h^l(\mathbf{x}^{(p)}, \mathbf{x}^{(p)})} = 1$$

By dividing both sides of the equation for the m-dependent sparse pre-activation kernel (Eqn. 25) with $\sqrt{K_h^l(\mathbf{x}^{(p)}, \mathbf{x}^{(p)}) K_h^l(\mathbf{x}^{(q)}, \mathbf{x}^{(q)})}$ and setting $p = q$, we get the following equation:

$$\frac{K_h^{l+1}(\mathbf{x}^{(p)}, \mathbf{x}^{(p)})}{K_h^l(\mathbf{x}^{(p)}, \mathbf{x}^{(p)})} = \frac{\sigma^{*2}}{2\pi} \left[ \left( 2I\left(0 \mid \tau\right) - 2\tau\sqrt{2\pi} \right) + m \left( 2I\left(\frac{\pi}{2} \mid \tau\right) - \tau\sqrt{2\pi} \right) \right]$$

Since we require the above to equal to 1, we have

$$\frac{\sigma^{*2}}{2\pi} \left[ \left( 2I\left(0 \mid \tau\right) - 2\tau\sqrt{2\pi} \right) + m \left( 2I\left(\frac{\pi}{2} \mid \tau\right) - \tau\sqrt{2\pi} \right) \right] = 1$$

By rearranging the terms, we obtain

$$\sigma^* = \sqrt{\frac{2\pi}{2I\left(0 \mid \tau\right) - 2\tau\sqrt{2\pi} + m \left( 2I\left(\frac{\pi}{2} \mid \tau\right) - \tau\sqrt{2\pi} \right)}} \tag{26}$$

or equivalently,

$$\sigma^* = \sqrt{\frac{2\pi}{g\left(0, f\right) + m g\left(\frac{\pi}{2}, f\right)}}$$

## D. Derivation of $m^*$

By dividing both sides of the equation for the m-dependent sparse pre-activation kernel (Eqn. 25) with $\sqrt{K_h^l(\mathbf{x}^{(p)}, \mathbf{x}^{(p)}) K_h^l(\mathbf{x}^{(q)}, \mathbf{x}^{(q)})}$ , we have

$$\frac{K_h^{l+1}(\mathbf{x}^{(p)}, \mathbf{x}^{(q)})}{\sqrt{K_h^l(\mathbf{x}^{(p)}, \mathbf{x}^{(p)}) K_h^l(\mathbf{x}^{(q)}, \mathbf{x}^{(q)})}} =$$

$$\sigma^2 \frac{1}{2\pi} \left[ \left( 2I\left(\theta^l \mid \tau\right) - \tau\sqrt{2\pi}(1 + \cos\theta^l) \right) + m \left( 2I\left(\frac{\pi}{2} \mid \tau\right) - \tau\sqrt{2\pi} \right) \right] \tag{27}$$

Since for the $\sigma^*$ mentioned above, $K_h^l(\mathbf{x}^{(q)}, \mathbf{x}^{(q)}) = K_h^{l+1}(\mathbf{x}^{(q)}, \mathbf{x}^{(q)})$, and the L.H.S. of the above equation becomes

$$c^{l+1} := \frac{K_h^{l+1}(\underline{x}^{(p)}, \underline{x}^{(q)})}{\sqrt{K_h^{l+1}(\underline{x}^{(p)}, \underline{x}^{(p)}) K_h^{+1}(\underline{x}^{(q)}, \underline{x}^{(q)})}} = \cos\theta^{l+1}$$

Therefore, the normalized kernel formula is

$$c^{l+1} = \frac{\sigma^{*2}}{2\pi} \left[ \left( 2I\left(\arccos c^l \mid \tau\right) - \tau\sqrt{2\pi}(1 + c^l) \right) + m \left( 2I\left(\frac{\pi}{2} \mid \tau\right) - \tau\sqrt{2\pi} \right) \right]$$

Viewing $c^{l+1}$ as a function of $c^l$, the slope of $c^{l+1}$ at $c^l = 1$ determines the disorderedness and orderedness of the kernel as discussed in the main text. If the slope equals 1, the kernel is determined

to be on the phase boundary. Here, we obtain the formula for $m^*$ that results in a kernel that exists on the phase boundary. The condition is formally written as

$$\frac{dc^{l+1}}{dc^l}|_{c^l=1} = 1$$

The derivative is given by

$$\frac{dc^{l+1}}{dc^l}|_{c^l=1} = \sigma^{*2}\frac{1}{2\pi}\left(2\frac{d}{dc^l}I\left(\arccos c^l \mid \tau\right) - \tau\sqrt{2\pi}\right) = 1$$

By substituting $\sigma^*$ with its expression found earlier (Eqn 26), we obtain

$$m^* = \left[2\frac{d}{dc^l}I\left(\arccos c^l \mid \tau\right)|_{c^l=1} + \tau\sqrt{2\pi} - 2I\left(0 \mid \tau\right)\right]\left(2I\left(\frac{\pi}{2} \mid \tau\right) - \tau\sqrt{2\pi}\right)^{-1}$$

This is equivalent to the expression in the main text. Now we compute the derivative $\frac{d}{dc^l}I\left(\arccos c^l \mid \tau\right)|_{c^l=1}$.

Decompose $\frac{d}{dc^l}I\left(\arccos c^l \mid \tau\right) = A + B$, where $A = \int_0^{\frac{\pi-\theta}{2}}\frac{\partial}{\partial c^l}r(\phi_0,\theta)d\phi$

$$A = -\int_0^{\frac{\pi-\theta}{2}} 2\exp\left(-\frac{\tau^2}{2\sin^2(\phi_0)}\right)\frac{\cos(\phi_0+\theta)}{\sin\theta}\sin(\phi_0) + \tau\frac{\cos(\phi_0+\theta)}{\sin\theta}\sqrt{\frac{\pi}{2}}\text{erf}\left(\frac{\tau}{\sqrt{2}\sin(\phi_0)}\right)d\phi_0$$

Therefore,

$$B = \frac{1}{2\sin\theta}r(\frac{\pi-\theta}{2},\theta)$$

where $r(\phi_0,\theta)$ is the integrand of $I(\theta \mid \tau)$.

$$r(\phi_0,\theta) = 2\exp\left(-\frac{\tau^2}{2\sin^2(\phi_0)}\right)\sin\left(\phi_0+\theta\right)\sin(\phi_0)$$

$$+ \tau\left(\sin\left(\phi_0+\theta\right)+\sin(\phi_0)\right)\sqrt{\frac{\pi}{2}}\text{erf}\left(\frac{\tau}{\sqrt{2}\sin(\phi_0)}\right) \quad (28)$$

## E. More experimental results

The numerical experiments on the generalization performance of the sparse and correlated weights NNGP kernels with $L = 2$ are presented in Figure S1,S2,S3. The kernels are trained on different numbers of training set samples and tested on test set sizes twice that of the training sets. The experiment was repeated over 8 trials for randomly selected training samples. The details of the figure layout and formatting are the same as that of the main text Figure 2. The improvement in the generalization performance by the weight correlation in the sparse regime is observed in all training examples. The benefit of having correlated weights diminishes with the increase in training set size ($P$) in the two-hidden layer architecture. Intuitively, having correlated weights is similar to having fewer parameters in a network, which in turn regularizes the network output yielding smoother data interpolation. This explains the result in the main text that the weight correlation decreases kernel dimensionality, i.e. increases the kernel width. With more training data, the benefit of this regularization diminishes.

The improvement from a weight correlation is more pronounced in deeper networks. Figure S4 shows that for a deeper network the improvement from weight correlation is consistently pronounced in the range of $P$ between 128 and 4096.

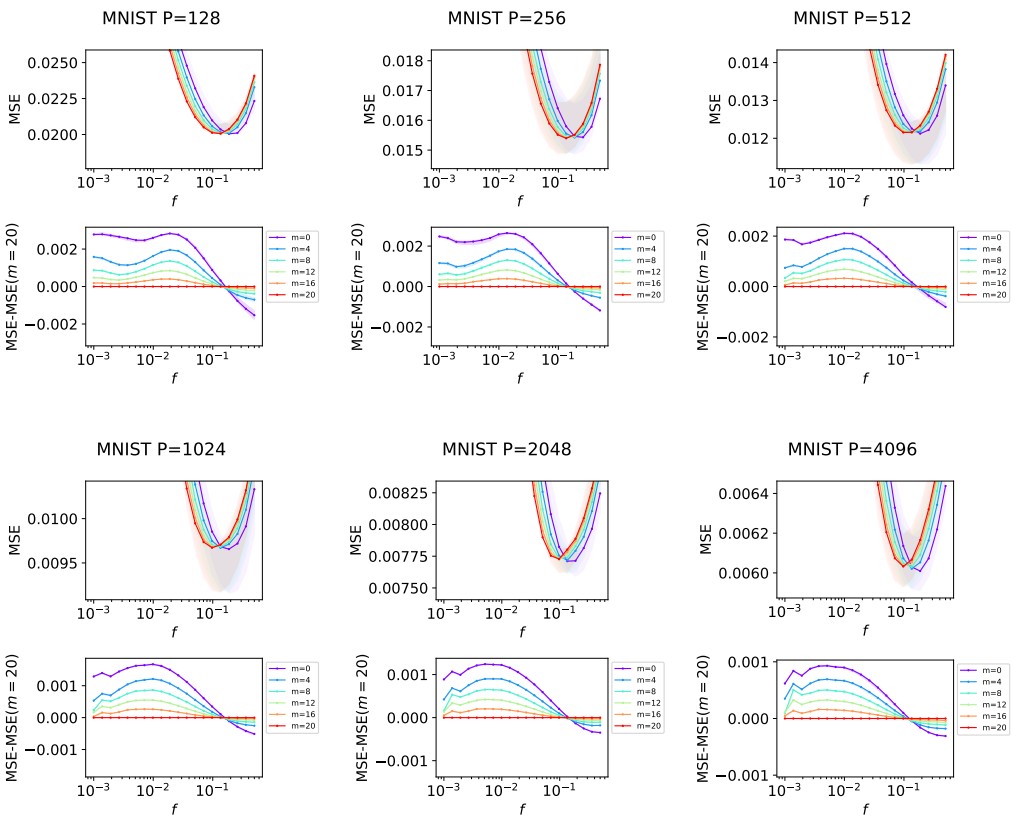

Figure S1: Experimental results on MNIST dataset of different training set sizes for $L = 2$.

## F. More comparison between the experiments and generalization theory

To show that the generalization theory predicts the numerically observed improvements due to weight correlation in the sparse regime, we compare the prediction by the theory and numerical experiments for different datasets (FigureS5). These figures serve the same purpose as Figure 3ab: to show that it is valid to utilize the generalization theory by Canatar et al. to understand the observed effect of correlated weights.

## G. Performances of deep NNGP models

Here we present the generalization performances of deep NNGP models with sparsity and correlated weights. We show that the optimal correlation level $m^*(f)$ given by the phase boundary gives the best-performing kernels at deeper depths, but not necessarily at shallower depths (Figure S6). That is because the kernels are far from the equilibrium points at shallower depths.

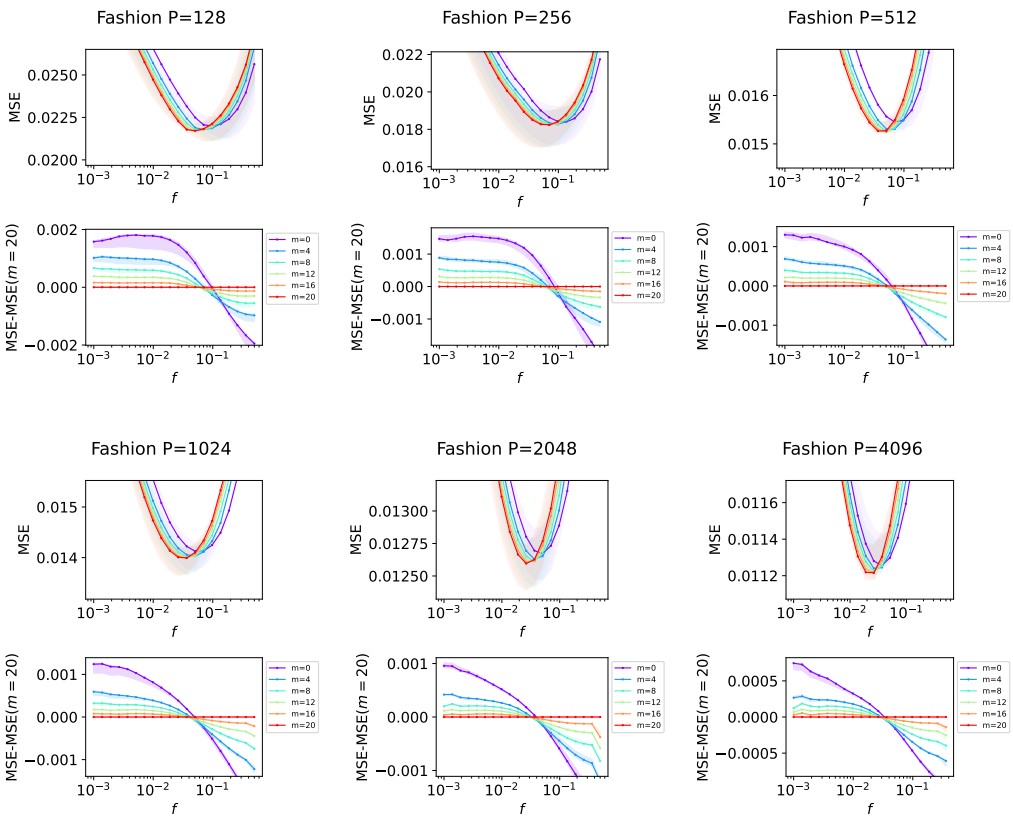

Figure S2: Experimental results on Fashion-MNIST dataset of different training set sizes for $L = 2$.

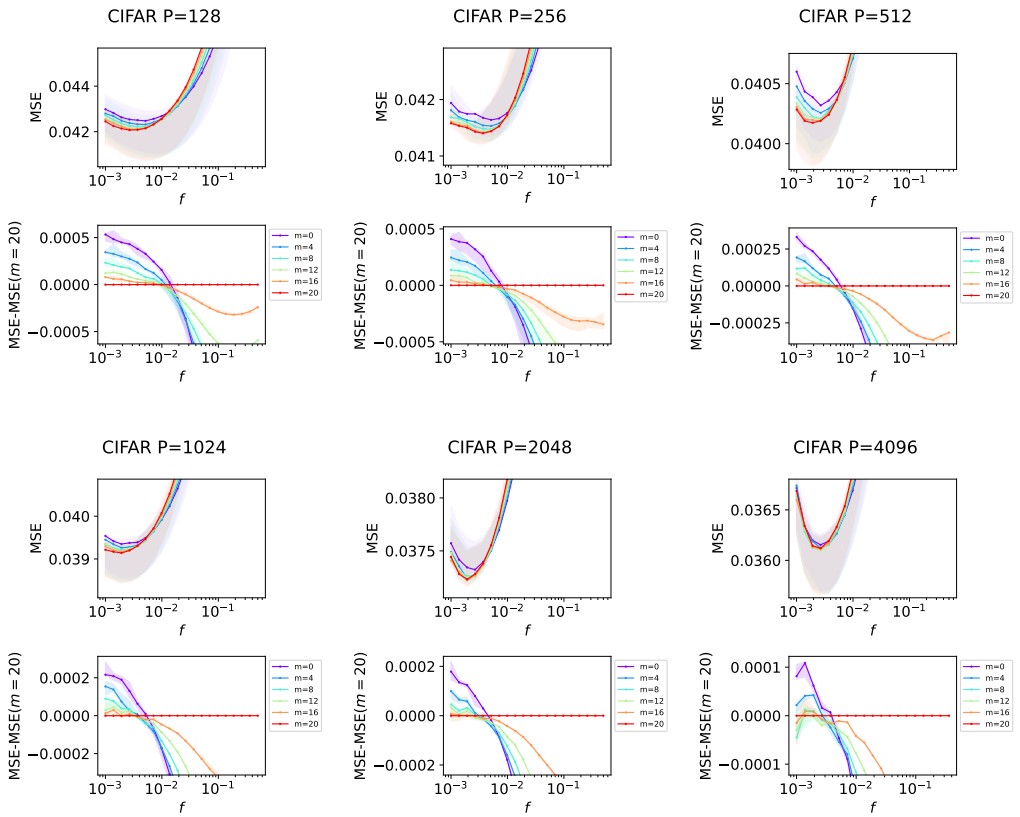

Figure S3: Experimental results on CIFAR10 dataset of different training set sizes for $L = 2$.

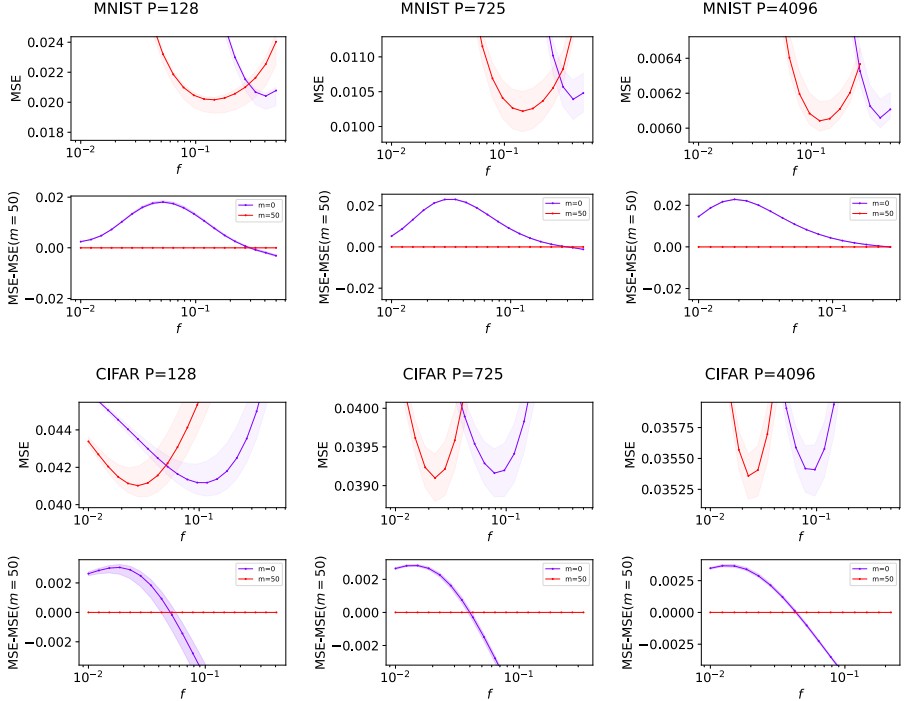

Figure S4: Experimental results on MNIST and CIFAR10 dataset of different training set sizes for $L = 4$.

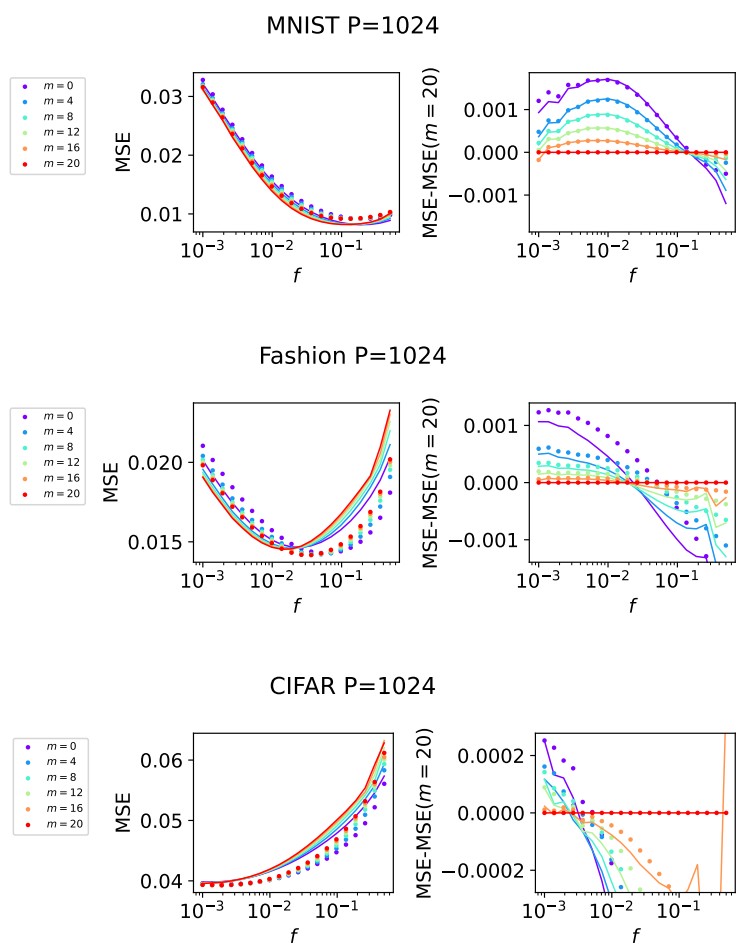

Figure S5: Comparison between the experimental results and theoretical predictions on generalization performances. Dotted markers indicate the experimental results. Lines indicate the theoretical results

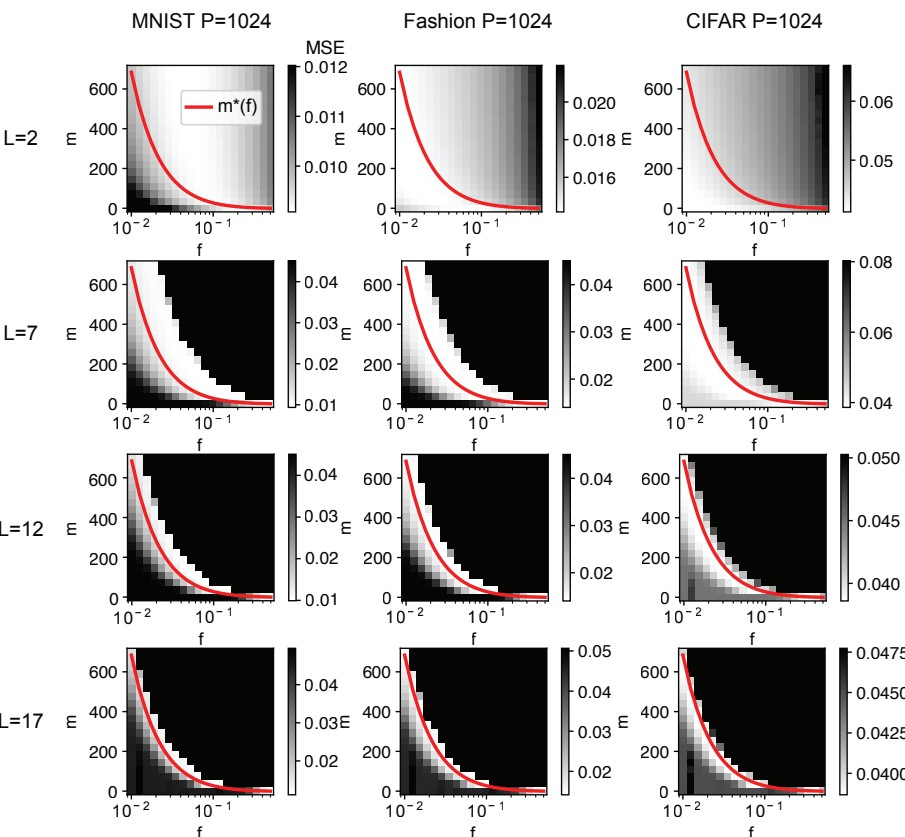

Figure S6: Empirical generalization performances of the kernels of different depths. $L$ is the number of hidden layers. Red line is the phase boundary that determines the theoretical optimal weight correlation degree $m^*$ as the function sparsity $f$.

