# OpenReview forum: "Sparse Activations with Correlated Weights in Cortex-Inspired Neural Networks"
_CPAL.cc/2024/Conference — CPAL 2024 (Proceedings Track) Oral_

### Official Review · Reviewer_FTVf · 2023-10-06
**Counteracting sparsity with weak correlations**

**Rating:** 7
**Confidence:** 1

**Review:**

### Summary :
This work suggests weakly correlated weights in the neural network Gaussian Process model enable learning useful sparse representations. Empirical evaluation demonstrates that weak correlations (via an intermediate convolutional layer) induce low-dimensional kernel Gram matrix in the sparse regime. Theoretical justification is also provided for the same observation which in turn predicts good generalization behaviour. Informally sparsity dissimilates the neural representations across layers while correlations counter this effect. For any sparsity level f, the authors derive an optimal correlation level that balances the tradeoff in generalization.

---

### Official Review · Reviewer_qzTS · 2023-10-07
**Review of Sparse Activations with Correlated Weights in Cortex-Inspired Neural Networks**

**Rating:** 7
**Confidence:** 3

**Review:**

### Summary of contributions:
This work demonstrates, both theoretically and empirically, how adding weak correlations between weights in an infinitely-wide neural network can counter the dissimilating effect of sparse activations which in turn improves generalization performance on three small scale datasets.

### Strengths:
1. This is a timely and important topic as sparse activations are observed in many neural network architectures and in particular the now ubiquitous transformer architecture.
2. The paper is well motivated based on prior art and biologically similar mechanisms within cortical circuits.
3. The study of using correlated weights in a NNGP setting appears to be novel.
4. The formula presented for calculating the optimal weight correlation is of particular interest. Future work exploring applications of correlated weights may be able to realize modest generalization performance improvements by implementing correlated weights.
5. Empirical results generally match very closely with results predicted by theoretical results, particularly for experiments with smaller training sets.

### Weaknesses:
1. Experiments with m=20 do appear to consistently yield the best generalization performance, especially at moderate to high sparsities with small training datasets (P<=512). However, as P is increased, the gap between m=0 and m=20 appears to narrow, particularly for CIFAR and MNIST. I also note that the gap between theoretical predictions and empirical results in Figure S4 appears to increase proportionally with P. A more detailed discussion on the effect of increasing training set size would improve the paper.
3. While typical of the NNGP literature, the datasets used for the analysis are small and the training set sizes used are very small for typical training of neural networks. It is unclear if the weight correlation benefits will remain if larger, real-world datasets are used. Did the authors conduct any experiments using training sets larger than 4096? If not, what was the reason for excluding larger numbers of training samples? I remain skeptical if benefits observed with weight correlation will continue to be present as the training set size is scaled up.
4. The empirical study is limited to the NNGP setting. Experiments with finite-width networks in addition to the kernel method experiments would help establish whether the proposed weight correlation methodology is broadly applicable to real-world neural networks.
5. For MNIST and FashionMNIST, the benefits of weight correlation appear to be most prominent in the highly sparse regime. However, the best absolute generalization performance is typically found at modest sparsities where the benefits of using correlated weights are less compelling.
6. Only a single type of network architecture was considered for the empirical study. Additional experiments with deeper models would improve confidence in the results.


### Clarifications:
1. "More recently, the presence of sparse activation has been observed in high-performance neural networks such as AlexNet, **ImageNet**, LeNet, and various models of Transformers, even without explicit regularization for sparsity."ImageNet" typically refers to the ILSVR challenge datasets and not a neural network architecture. Please confirm intended network.
2. "f" in equation 5 refers to the "sparsity level". However, I believe the actual intent of this variable is to represent the "fixed fraction of neurons with non-zero activations" as per [1]. I would expect sparsity level to be defined as (1-f) if this is correct. Please clarify.
3. Figure 3 caption states that "Red triangle marks f = 0.2, m = 20 as the best-performing model.". This seems like a reach to me as essentially all correlation levels have the same performance at that sparsity. What is the absolute difference between m=0 and m=20 at the red triangle? If it's as small as it appears in the plot, I recommend removing this statement as it seems disingenuous.
4. On line 192, P is defined as "the size of the training set". This is clearly established in plots throughout the paper, but seems to be placed somewhat awkwardly with respect to equation 13 since P does not appear in that equation.

### Suggested changes:
1. Adding hyperlinks to the in-text citations that link to the bibliography would be helpful.
2. Move definition of P to a more appropriate location, preferably near Figure 2 when it is first used in caption and figure plot titles.
3. In Figure 4b), m=150 series is hard to read. Consider use of a different color.

### Broader impact concerns:
None.

### Citations
[1] C. Chun and D. D. Lee, “Sparsity-depth Tradeoff in Infinitely Wide Deep Neural Networks.” arXiv, May 17, 2023. doi: 10.48550/arXiv.2305.10550.

---

### Official Review · Reviewer_sLsf · 2023-10-08
**An interesting work discovering the correlation between weight correlation and activation sparsity within infinitely-wide networks.**

**Rating:** 6
**Confidence:** 3

**Review:**

This paper focuses on the benefits of sparse activations in infinitely-wide networks and studies how weak correlations in the weights can improve the generalization performance. The weights have been typically assumed to be independent so understanding the effect of correlated weights is currently limited. To this end, this paper first proposed a variant of NNGP for correlated weights, showing that the inducing of correlated wights is capable of improving the generalization performance in the sparse regime. The corresponding theoretical explanation is further provided by extending the recent advances in generalization theory. Eventually, the optimal weight correlation give a target sparsity has been introduced, improving the practical usage of this paper.

Overall, this paper is presented with high quality and clarity. I enjoy much when reading the introduction and review of sparse NNGP.

The novelty of this paper mainly lies on (1) the formulation on extending of NNGP for correlated weights; (2) the empirical evidence for the benefits of weights correlation to feature sparsity, and to generalization performance; (3) Theoretical proof.

My major concern is why this paper mainly focus on random weighted sparse networks? Can the findings be generalized to trained sparse networks, the most closely one is "The lazy neuron phenomenon: On emergence
318 of activation sparsity in transformers"; and sparse training regime i.e., SET (https://www.nature.com/articles/s41467-018-04316-3), SNIP (https://arxiv.org/abs/1810.02340), ITOP (https://arxiv.org/abs/2102.02887).

I also list several minor cons of this manuscript here:

(1) The authors mention that "Therefore, sparse random models do not perform well with deep architectures [12]." However, IMHO, there are empirical papers showing that when models get deeper and wider, sparse random models actually perform better, i.e., random pruning https://arxiv.org/abs/2202.02643, and random sparse GNNs: https://arxiv.org/abs/2211.15335. While the randomness in this manuscript is slightly different from the papers I mentioned, it is necessary to at least discuss and explain this two seemingly counter-arguments.

(2) I encourage the authors provide a contribution summary in the early of the paper to improve the readiness of this paper.

(3) When this paper mentions in the intro "Currently, there is no theoretical explanation of how deep network models such as the Transformers benefit from sparsity" as their motivation, I expect to see any analysis of Transformers in the main paper. However, it is not presented. I encourage the authors to say something about Transformers or simply remove this statement.

---

### Meta-Review · Area_Chair_C2co · 2023-11-09

**Recommendation:** Accept (Oral)
**Confidence:** 4

**Metareview:**

The paper investigates the role of sparsity in infinitely-wide Neural Network Gaussian Process (NNGP) models, particularly focusing on the benefits of correlated weights on generalization performance. Strengths highlighted by reviewers include the theoretical advancements in NNGP by looking at correlated weights rather than independent weights, and the proposed optimal weight correlation formulation for specific sparsity targets. Some reviewers expressed concerns regarding the limited scope of experiments, primarily relying on small datasets and training set sizes, and questioned whether the authors insights for random sparse networks carry over to trained sparse networks. However, these issues were addressed in the author rebuttal to the apparent satisfactions of the reviewers. Overall, a strong theoretical work that should have broad appeal to researchers interested in understanding the role sparsity in both biological and artificial neural networks.

---

### Decision · Program_Chairs · 2023-11-19

**Decision:**

Accept (Oral)

**Comment:**

This paper investigates the impact of sparse activations and correlated weights in infinitely-wide Neural Network Gaussian Process (NNGP) models, aiming to improve generalization performance. Reviewers appreciate the theoretical advancements in NNGP related to correlated weights and the formulation of optimal weight correlation for specific sparsity targets. However, some concerns include the limited experimental scope, reliance on small datasets and training set sizes, and questions about the applicability of findings to trained sparse networks. Nevertheless, these concerns were addressed in the author rebuttal. Overall, the paper presents a strong theoretical contribution relevant to researchers interested in sparsity in neural networks, both biological and artificial.

The action PC chair for this paper is Atlas Wang, who made the decision after carefully reading the paper as well as the comments by all reviewers and AC. The decision is agreed by all PC chairs.